psychology/cognition

language production, joint action, monitoring, Stroop

**Author for correspondence:**
Chiara Gambi
e-mail: gambic@cardiff.ac.uk

# Interference in the shared-Stroop task: a comparison of self- and other-monitoring

Martin J. Pickering[1], Janet F. McLean[2] and
Chiara Gambi[3]

[1]Department of Psychology, University of Edinburgh, 7 George Square, Edinburgh EH8 9JZ, Scotland, UK
[2]School of Applied Sciences, Abertay University, Dundee DD1 1HG, Scotland, UK
[3]School of Psychology, Cardiff University, 70 Park Place, Cardiff CF10 3AT, Wales, UK

MJP, 0000-0002-2005-049X; CG, 0000-0002-1568-7779

Co-actors represent and integrate each other's actions, even when they need not monitor one another. However, monitoring is important for successful interactions, particularly those involving language, and monitoring others' utterances probably relies on similar mechanisms as monitoring one's own. We investigated the effect of monitoring on the integration of self- and other-generated utterances in the shared-Stroop task. In a solo version of the Stroop task (with a single participant responding to all stimuli; Experiment 1), participants named the ink colour of mismatching colour words (incongruent stimuli) more slowly than matching colour words (congruent). In the shared-Stroop task, one participant named the ink colour of words in one colour (e.g. red), while ignoring stimuli in the other colour (e.g. green); the other participant either named the other ink colour or did not respond. Crucially, participants either provided feedback about the correctness of their partner's response (Experiment 3) or did not (Experiment 2). Interference was greater when both participants responded than when they did not, but only when their partners provided feedback. We argue that feedback increased interference because monitoring one's partner enhanced representations of the partner's target utterance, which in turn interfered with self-monitoring of the participant's own utterance.

## 1. Introduction

People are remarkably adept at performing joint actions, such as playing a duet, ballroom dancing or holding a conversation. When performing these actions, it is not enough for them to be successful on their own; they must also perform their action so that it is compatible with their partner's action. To do this, they

must be able to predict and monitor each other's actions and to use information about their partner's actions in preparing their own actions. A large body of research on joint spatial action tasks has suggested that people are able to construct representations of their partner's action as well as their own, and moreover, that self- and other-representations interact and affect one another (e.g. [1–3]), in a way that suggests each of them has constructed a representation of their joint task as well as of their respective individual tasks.

There is increasing evidence that people also construct representations of their partner's utterances in joint language tasks [4–10]. But it is currently unclear to what extent representations of others' utterances and representation of one's own utterances are integrated and affect one another. Specifically, some studies suggest that representations of others' utterances have effects that are analogous to those of representations of one's own utterances (e.g. interacting in similar ways with other linguistic representations or eliciting similar brain responses; [4,6,10]), but other work suggests that representation of others' utterances are not as tightly integrated with representations of one's own utterances [5,7–9]. We ask whether speakers are more likely to tightly integrate self- and other-representations when they have to monitor their partner's utterances for correctness. Other-monitoring—that is monitoring of the utterances spoken by another speaker—is of course an important component of comprehension in general [11–13]. However, it might be particularly important during dialogue, when monitoring the utterances of one's interlocutor is critical not only for comprehension (of those utterances) but also for checking the interlocutor's understanding of one's own utterances. In fact, in dialogue, other-monitoring might also be tightly integrated with self-monitoring [11,14,15]—that is, monitoring of one's own utterances before and after they are spoken [11,16]. Thus, we tested whether encouraging participants to monitor their partner's utterances—because they were required to provide feedback to their partner—in a joint language task would make them more likely to tightly integrate self- and other-representations.

Importantly, joint spatial action tasks do not only show that participants represent their partner's actions in a way that affects their own actions, but also that such effects tend to be stronger when the partner's actions are (i) more salient and when (ii) there is a closer relationship between the participants. Most of the evidence for these conclusions comes from variations of the so-called joint Simon task, which was introduced by Sebanz *et al.* [17]. They had participants perform a spatial compatibility (Simon) task, in which they were presented with a finger wearing a red or green ring that pointed left or right. Participants in the solo task (i.e. the classic Simon task) responded to red stimuli by pressing a left button and to green stimuli by pressing a right button (solo task), and were faster when the finger pointed toward the button that they had to press than to the other button. Crucially, when participants took part in pairs and one participant responded to (say) red stimuli but the other participant did not respond, there was no spatial compatibility effect (individual task). By contrast, when one participant in the pair responded to red stimuli and the other to green stimuli, the compatibility effect returned (joint task).

This joint Simon effect occurs when a salient stimulus provides a frame of reference for the participant's own action (i.e. the action is coded as being 'left' or 'right' in relation to this stimulus [18], see also [19]). Accordingly, the effect is larger when participants regard the other responding hand as being more 'separate' from their own hand [18]. Further, while non-social salient stimuli can elicit the joint Simon effect (e.g. a Chinese waving cat; [20]), both the occurrence and the magnitude of joint spatial effects is modulated by co-presence and by the social relationship between participants. Although joint spatial effects do occur when no co-actor is present [21,22], when each participant cannot see or hear the other [23] and when participants sit close together but do not collaborate [19], they appear to be less reliable: some studies have reported no effects when the partner is not visible [24] or a non-biological entity [25]. Finally, stronger effects occur with likeable than intimidating partners [26]. Thus, taken together, joint spatial action studies suggest that when the relationship between participants is close (either as a result of task structure or social factors), other-representations are enhanced—that is, participants are more likely to represent their partner's actions as well as their own, and stronger effects ensue.

But to what extent do these conclusions generalize to non-spatial joint activity? A good example is interactive linguistic communication, in which speakers respond appropriately and rapidly to their partner's contributions [27]. For example, the gaps between contributions to dialogue tend to be extremely short (often around 200 ms; [28]). To respond appropriately, interlocutors probably predict their own and their partner's utterances. Pickering & Garrod [29] specifically proposed that interlocutors engage in covert production of one another's utterances, so that each represents the perceived utterance of his or her interlocutor using some of the same mechanisms used when representing utterances he or she is about to produce (see also [30,31]).

When people use language, they intend to communicate, and assume that their partners intend to understand what they are saying [32]. We might therefore expect joint-representation effects in non-spatial tasks involving language. Indeed, similar to joint action tasks, participants in joint language tasks represent their partner's task and actions (utterances) as well as their own, though the degree to which self- and other-representations are integrated varies. For example, Gambi *et al*. [8] asked pairs of participants to name pairs of pictures superimposed on each other; participants sat in different rooms and did not interact, though they were visible to each other in their peripheral vision. Participants' naming latencies were affected by their beliefs about their partner's task: speakers were slower at naming pairs of pictures when they believed that their partner was also naming but not when they believed that their partner was silent or performing a different task on the pictures. However, it did not matter whether participants believed their partner was naming the two pictures in the same or different order to themselves. Therefore, Gambi *et al*. argued that, in joint language tasks, speakers represent that their interlocutor is preparing to speak but not what he or she is preparing to say, indicating only partial integration of self- and other-representations. By contrast, when participants take turns speaking with a co-present partner, they may additionally represent aspects of the linguistic content of the partner's utterance. Kuhlen & Abdel Rahman [10] showed that semantic interference accumulates not only as a function of how many related pictures the speaker has previously named, but also as a function of how many related pictures were previously named by a partner (whether or not this partner was audible, but only if the partner was seated in the same room [33]; see also [9]). Similarly, in an EEG study where participants took turns naming pictures with a co-present confederate [4], speakers showed an effect of lexical frequency when their partner was about to name the pictures, and not just when they were about to name the pictures themselves. They therefore appeared to (covertly) engage in lexical processing when their co-present partner was naming even if they were not overtly naming themselves. Taken together, this evidence from joint language tasks suggests that joint representations are formed but also that their nature and strength may well depend on aspects of the social context (e.g. co-presence; [34]).

No study so far has manipulated whether or not participants were requested to provide feedback to their partners as to the correctness of their utterances. While one previous study did ask participants to provide feedback to each other [6], it did not include a condition without feedback, so the data do not speak directly to the question that we ask in the current paper. However, it employed a similar task and set-up to the current study, so below we describe it and its findings in some detail.

Demiral *et al*. [6] had participants perform a delayed go-no-go version of the Stroop task, where participants responded on trials where the word was printed in one ink colour (e.g. green; go trials) but not on trials where the word was printed in a different ink colour (e.g. red; no-go trials). The classic Stroop effect refers to the finding that people experience difficulty naming the ink colour of a word if that word's meaning is incompatible with the colour of the word (see [35]). In the joint version of the task, each participant named words of a particular colour (e.g. red), while ignoring words of the other colour (e.g. green), and provided feedback to their partner's utterance ('yes' if their partner's response was correct, 'no' if it was not). In the individual version of the task, only one participant named words of a particular colour (e.g. red) and their partner provided feedback. Compared with the individual task, EEG data showed an increased P3b (i.e. a positive deflection peaking roughly 300 ms after stimulus onset) on no-go trials during the joint task, suggesting participants mapped the stimulus onto their partner's upcoming response when it was their partner's turn to respond more than when it was nobody's turn to respond (i.e. in the individual task). This finding is compatible with the proposal that participants form representations of others' utterances when they are explicitly required to monitor such utterances, though since Demiral *et al*. did not run a version of the joint task without feedback, it is unclear whether differences between the individual and joint tasks occurred because the partner was naming the other colour, or specifically because the participant had to monitor what the partner was saying.

In favour of the latter possibility, in a go-no-go version of the Stroop task that did not include a monitoring task, Saunders *et al*. [36] found comparable levels of interference whether or not the participants were sharing the task with a human partner: similar levels of interference were induced by colour words that corresponded to (i) another ink colour assigned to the same participant and to (ii) ink colours assigned to a task partner (relative to ink colours that were not assigned to either the participant or the task partner), but this was the case even when there was in fact no partner, suggesting that Stroop interference may arise fairly automatically from reading colour words and may not be dependent on social factors. Taken together, the findings of Demiral *et al*. and Saunders *et al*. thus suggest that an explicit monitoring task may be necessary to elicit a joint Stroop interference effect. We tested this claim in the current study.

But before we describe the current study's rationale and design, it is interesting to consider another finding from Demiral *et al*. [6]. In that study, there was no evidence of increased Stroop interference in the joint compared with the individual version of the task. In fact, it revealed a *reduced* congruency effect on the N2 component—indexing perceptual conflict [37]—in the joint compared with the individual task. Note that this study did not analyse response times, as responses were delayed to avoid speech artefacts in the EEG. Nevertheless, this finding suggests that representing a co-actor's utterance may not only cause additional interference between competing response alternatives (as indicated by the P3b findings described above), but also attenuate perceptual conflict.

This hypothesis is consistent with the few studies that compared a joint and an individual version of the picture-word interference (PWI) task. In the PWI task, participants name pictures while ignoring superimposed distractor words and are typically slower when distractors are semantically related to the pictures than when they are unrelated (i.e. a semantic interference effect; [38]). For example, Sellaro *et al*. [39] found a reduced semantic interference effect in a condition in which participants named pictures and were (falsely) told they had a partner in another room who read the superimposed distractor words (see also [40]). Similarly, Kuhlen & Abdel Rahman [41] found that when the PWI task is embedded in a communicative game, with one participant naming the distractor words and the other, co-present participant naming the pictures, semantic interference is greatly reduced (compared with a non-communicative, standard version of the PWI task). A possible reason is that naming pictures in a communicative setting enhances semantic facilitation at the conceptual level (due to distractor and target belonging to the same semantic category). While findings from joint PWI tasks may not translate directly to joint Stroop tasks because of several important methodological differences (e.g. in Stroop, the task-irrelevant stimulus is spatially co-located with the task-relevant stimulus and strongly activates the interfering response), evidence for reduced perceptual conflict in the joint task of Demiral *et al*. [6] and evidence for reduced interference in joint PWI tasks [39–41] means it is unclear whether one would expect joint Stroop interference to increase or decrease in a joint compared with an individual go-no-go version of the task.

In sum, more evidence is needed to shed light on the varying contribution of task and social factors on joint linguistic actions. Specifically, it may be that representation of others' utterances in a more interactive situation, such as one requiring monitoring of each other's utterances, is enhanced compared with a less interactive situation in which one partner's utterances can easily be ignored. Pickering & Garrod [11] proposed that other-monitoring and self-monitoring rely on similar mechanisms: in either case, the monitoring partner builds a (predictive) representation of either his or her own utterance or the other's utterance, which is used to rapidly compare the expected utterance with the utterance that is actually produced, and either correct errors (in self-monitoring) or flag up a lack of understanding (in other-monitoring). If this proposal is correct, assigning participants the task of monitoring their partner's utterances should lead them to construct representations for those utterances using the same mechanisms that are responsible for constructing representations of their own utterances during self-monitoring (i.e. when they name the ink colour during the Stroop task). Thus, monitoring should enhance the likelihood of the other's utterances affecting the participant's production of his or her own utterances. Note that, while these predictions stem naturally from Pickering and Garrod's account, they may also be explained by other-monitoring accounts in which processes and representations are shared between production and comprehension [42]. We return to this point in the General discussion.

To investigate these issues in a tightly controlled manner, we used the Stroop task. Much work has used the Stroop task in a social context ([43,44], e.g. [45]). This work shows that the interference effect tends to be reduced when a passive bystander is present in the room with the participant (so-called 'social facilitation'; [46]). The underlying mechanisms of this effect are disputed, but importantly we controlled for it in this study by comparing conditions in which another person was present, but inactive (single-response task) with conditions in which another person was present and responded on trials that were no-go trials for the participant (joint-response task).

The version of the Stroop task used in our experiments had four stimuli, and participants were instructed to respond as soon as a stimulus appeared on the screen. In the congruent condition, the word *red* was printed in red and the word *green* was printed in green; in the incongruent condition, the word *red* was printed in green and the word *green* was printed in red. When a single participant named the colour of the word, we predicted that he or she would take longer (and make more errors) naming incongruent than congruent colours.

But what happens when two participants share the task, with one naming words in the colour red and the other naming words in the colour green? Can the mere fact that incongruent colour words

evoke the other person's response induce a larger Stroop interference effect compared with when there is no such response? Or is mutual feedback necessary to achieve this result? Research indicates that speakers are affected by appropriate feedback on their utterance (e.g. narrators tell better stories to responsive than unresponsive addressees; [47]), but we do not know whether the experience of providing feedback also affects listeners' subsequent utterances (i.e. when they become speakers again). While we do not know the exact mechanism by which receiving (or providing) feedback affects language processing, it is likely that introducing a requirement to provide feedback will make participants more likely to closely monitor their partner's utterances.

Experiment 1 was a traditional Stroop experiment using a single participant. We expected to observe a large Stroop interference effect in this experiment. But in Experiments 2 and 3, participants took part in pairs. One participant named the colour of words in one colour (e.g. red) while ignoring words in the other colour (e.g. green). We expected the magnitude of the Stroop interference effect to be reduced in Experiments 2 and 3 compared with Experiment 1, for two reasons. First, participants were now assigned only one overt response, rather than two, which could reduce the amount of interference at the response selection stage. Second, participants now performed the task in the presence of another person, which could also reduce interference due to the faciliatory effect of social contexts (e.g. [45]). The important comparison, though, is within-experiments. In both Experiments 2 and 3, we manipulated whether the participant's partner named the colour of words in the other colour or did not. If speakers construct representations of their partner's utterances as well as of their own utterances and integrate them into a representation of the joint task [29,31], other-representations may interfere with selection of the correct response on go trials. Speakers may construct such other-representations when their partner responds to stimuli in the other colour but should not construct such representations when their partner does not. In other words, speakers may experience more interference when each partner performs 'half' of the Stroop task than when one partner performs 'half' of the task and the other partner does not perform the task (but see [36]).

Crucially, we also manipulated whether the addressee provided feedback to the speaker (Experiment 3) or did not (Experiment 2). The feedback consisted of 'yes' when the speaker produced the right colour name and 'no' otherwise. Without feedback, participants could simply perform their own task and overhear their partner's activity; with feedback, participants are engaged in the task of other-monitoring. We hypothesize that other-monitoring requires building a representation of the other's utterance and comparing this representation to what they actually say. If so, other-monitoring should lead to increased interference: as well as monitoring the other, the participant is also monitoring her own utterances, and confusion between the target for self- and other-monitoring could lead to an enhanced shared-Stroop effect in the presence of feedback but not otherwise.

# 2. Experiments

We conducted three experiments involving the same two-choice Stroop task but varied the participants' task. The stimuli (*red* in red, *green* in green, *red* in green and *green* in red) were presented in the centre of an 18-inch colour monitor in lower case Arial typeface. Stimuli presentation and data recording were operated by DMDX [48], through a PC computer in a Psychology testing cubicle. Single participants sat centrally in front of the screen; pairs of participants sat side-by-side approximately 30 cm apart. Before each experiment, there were 32 practice trials. Participants first stated the colour of eight red and eight green squares presented in a random order, and then read aloud eight instances of the word *red* and eight of the word *green* (in black font) in a random order. The experimenter was present during the practice trials, but outside the cubicle during the experimental trials.

Each experiment consisted of three blocks of 128 trials with a short break between each block. The first eight trials were warm-up trials (two of each type). The remaining trials comprised 60 congruent trials (30 in each colour) and 60 incongruent trials (30 in each colour). The trials were presented on a white screen in a pseudo-random order with the constraint that there were no more than four consecutive trials of the same type. Each trial started with a fixation cross presented for 500 ms (centrally positioned), followed by a blank screen for 500 ms, and then the word. Participants were approximately 70 cm from the screen (with the word roughly at eye level).

Each word was displayed for 500 ms with a stimulus-onset asynchrony of 1500 ms. Participants were asked to identify the colour of the word as quickly and accurately as possible. Response times were the time taken to state the colour of the word from its onset. Each response was individually analysed using CheckVocal [49]. Response times (RTs) and accuracy were analysed using mixed-design ANOVAs, after

**Table 1.** Mean response times in milliseconds (with standard deviations in parentheses) and number of errors in Experiments 1–3. Note that RT means (and standard deviations) are for the trimmed data, and error totals do not include the removed outliers.

| experiment | condition | congruent | | incongruent | | RT |
|---|---|---|---|---|---|---|
| | | RT (ms) | errors | RT (ms) | errors | difference |
| one (solo) | - | 441 (100) | 23 | 502 (133) | 236 | 61 |
| two (pairs without feedback) | single-response | 403 (78) | 3 | 414 (85) | 12 | 11 |
| | joint-response | 401 (68) | 3 | 416 (79) | 16 | 15 |
| three (pairs with feedback) | single-response | 435 (80) | 2 | 447 (81) | 2 | 12 |
| | joint-response | 441 (100) | 15 | 474 (128) | 40 | 33 |

averaging over trials by participants. All the experimental materials, data and analyses are available at https://osf.io/hg3du/.

## 2.1. Experiment 1: solo task

Experiment 1 investigated Stroop interference in an individual, who responded to both red and green stimuli.

### 2.1.1. Participants

We did not conduct power calculations prior to data collection. The sample size for this study was determined on the basis of the extensive literature on the Stroop effect, suggesting that it is a robust effect with a large effect size (e.g. [50]). Twelve participants (10 females and 2 males) were paid to participate. In all experiments, the participants were native English speakers from the University of Edinburgh community and reported no reading difficulties or colour-vision problems.

## 2.2. Results

We analysed the response time data from correct responses (94%). To prepare the data for analyses, all RTs below 200 ms were removed (1 trial) before we conducted a recursive trimming procedure in which the criterion cut-off for outlier removal was established independently for each participant in each condition, by reference to the sample size in that condition [51]. In this way, we discarded 0.3% of the data (0.6% in the congruent condition, 0.0% in the incongruent condition). Table 1 shows the mean and standard deviation for the data included in the response time analyses and the error totals (after outliers were removed) that were subjected to the accuracy analyses. Participants named colours more quickly on congruent than incongruent trials, $t_{11} = -7.565$, $p < 0.001$, $d = 2.18$ and also more accurately, $t_{11} = -8.70$, $p < 0.001$, $d = 2.51$.

## 2.3. Experiment 2: pairs without feedback

Experiment 2 used pairs of participants, and we manipulated whether one participant responded to stimuli of one colour (red for half the participants, green for the other half) and the other did not respond (single-response condition), or whether one participant responded to red stimuli and the other responded to green stimuli (joint-response condition). If participants represented each other's potential responses, interference should be greater in the joint- than single-response condition.

### 2.3.1. Participants

We did not conduct power calculations prior to data collection. Since we expected Stroop interference to be reduced when participants only responded to words in one colour (because the interfering colour was not in their response set; see e.g. [52]) and in the presence of another person (e.g. [45]), we aimed to double the sample size used in Experiment 1. We thus tested 24 participants in each of the two conditions. In total, 72 participants (40 females, 32 males) were paid to participate and were assigned

to same-gender pairs. Twenty-four pairs were assigned to the single-response condition, and twelve pairs were assigned to the joint-response condition.

## 2.3.2. Results

The design yielded data for 24 single-response participants (either red or green) and 24 joint-response participants (both participants in each of the 12 joint pairs provided data, one red, one green). We analysed the response time data from correct responses (99.6%) and discarded 1.0% of the data following trimming (0.7% in the congruent condition, 1.3% in the incongruent condition), table 1.

*Response times.* A 2 (response condition, between-participants) × 2 (congruency, within-participants) ANOVA revealed an effect of congruency, $F_{1,46} = 52.3$, $p < 0.001$, $\eta_p^2 = 0.53$, with participants being slower to respond to incongruent than congruent words. There was no effect of response condition, $F_{1,46} = 0.00$, $p = 0.98$, $\eta_p^2 = 0.00$, and no interaction, $F_{1,46} = 1.84$, $p = 0.18$, $\eta_p^2 = 0.04$. Planned contrasts revealed a congruency effect in both the single-response condition, $t_{46} = -6.07$, $p < 0.001$, and the joint-response condition, $t_{46} = -4.16$, $p < 0.001$. Under conditions in which participants did not provide feedback to each other's responses, the (small) congruency effect occurred when one participant responded to words of one colour. However, the effect was not enhanced when the other participant responded to words of the other colour.

*Accuracy.* There were more errors in incongruent than congruent condition, $F_{1,46} = 9.75$, $p < 0.01$, $\eta_p^2 = 0.18$. There was no effect of response condition $F_{1,46} = 0.23$, $p = 0.63$, $\eta_p^2 = 0.01$ and no interaction, $F_{1,46} = 0.32$, $p = 0.57$, $\eta_p^2 = 0.01$. Planned contrasts revealed a congruency effect in the joint-response condition, $t_{46} = -2.61$, $p = 0.01$, but no effect in the single-response condition, $t_{46} = -1.81$, $p = 0.08$.

## 2.4. Experiment 3: pairs with feedback

Experiment 2 found no evidence that participants represented their partner's utterances as their own when they each simply responded to two of the four conditions. Although participants spoke in the presence of another person, they did not address that person or respond to their utterances. Participants may construct representations of another's utterances when they establish a closer relationship with their partners on the basis of a minimal form of interaction, where they have to closely monitor one another's utterances. We therefore conducted Experiment 3, which was identical to Experiment 2, except that the non-respondent participant provided feedback, by uttering 'yes' if the responder produced the right colour name and 'no' otherwise. Thus, in the single-response condition, one participant responded to stimuli of one colour and the other provided feedback to their partner's responses. In the joint-response condition, both participants responded to stimuli of one colour and provided feedback to their partner's responses.

### 2.4.1. Participants

We used the same sample size as in Experiment 2. Seventy-two further participants (54 females, 18 males) were paid to participate and were assigned to same-gender pairs, as in Experiment 2.

### 2.4.2. Results

We analysed the response time data from correct responses (98.8%). Three trials were removed as they were under 200 ms, and 1.2% of the data was discarded following trimming (1.1% in the congruent condition, 1.1% in the incongruent condition), table 1.

*Response times.* A 2 × 2 ANOVA revealed an effect of congruency, $F_{1,46} = 80.4$, $p < 0.001$, $\eta_p^2 = 0.64$, with participants being slower to respond to incongruent than congruent words. There was no effect of response condition, $F_{1,46} = 0.98$, $p = 0.33$, $\eta_p^2 = 0.02$, but there was an interaction, $F_{1,46} = 17.2$, $p < 0.001$, $\eta_p^2 = 0.27$, with the difference between congruent and incongruent trials being smaller in the single-response condition than the joint-response condition. Planned contrasts revealed a congruency effect in both the single-response condition, $t_{46} = -3.41$, $p < 0.001$, and the joint-response condition, $t_{46} = -9.28$, $p < 0.001$. Under conditions in which participants provided feedback to each other's responses, a (small) congruency effect occurred when one participant responded to words of one colour, but the effect was enhanced when the other participant responded to words of the other colour.

*Accuracy.* There were more errors in the incongruent than the congruent condition, $F_{1,46} = 7.78$, $p < 0.01$, $\eta_p^2 = 0.15$. However, there was also a main effect of response condition, $F_{1,46} = 13.24$, $p < 0.001$,

$\eta_p^2 = 0.22$, and an interaction, $F_{1,46} = 7.78$, $p < 0.01$, $\eta_p^2 = 0.15$. Planned contrasts revealed a congruency effect in joint-response condition, $t_{46} = -3.95$, $p < 0.001$, whereas in the single-response condition participants made the same (small) number of errors on congruent and incongruent trials.

## 2.5. Combined analysis of Experiments 2 and 3

*Response times.* A 2 (experiment, between-participants) × 2 (response condition, between-participants) × 2 (congruency, within-participants) ANOVA revealed effects of experiment, $F_{1,92} = 13.2$, $p < 0.001$, $\eta_p^2 = 0.13$, with participants responding faster in Experiment 2 (no feedback) than Experiment 3 (feedback), and congruency, $F_{1,92} = 132.4$, $p < 0.001$, $\eta_p^2 = 0.59$, with participants responding faster to congruent than incongruent words. There were also interactions between experiment and congruency, $F_{1,92} = 9.17$, $p < 0.01$, $\eta_p^2 = 0.09$, and response condition and congruency, $F_{1,92} = 17.3$, $p < 0.001$, $\eta_p^2 = 0.16$. More importantly, there was a three-way interaction, $F_{1,92} = 6.60$, $p < 0.05$, $\eta_p^2 = 0.07$, which indicated that the difference between the congruency effect in the single- and joint-response conditions was greater in Experiment 3 than Experiment 2. The presence of partner feedback therefore increased the congruency effect that occurred in the joint-response condition.

*Accuracy.* The overall number of errors did not differ significantly between experiments, $F_{1,92} = 2.36$, $p = 0.13$, $\eta_p^2 = 0.03$. Confirming separate analyses of Experiments 2 and 3, there were more errors in the incongruent than the congruent condition, $F_{1,92} = 17.3$, $p < 0.001$, $\eta_p^2 = 0.16$. As in Experiment 3, there was a main effect of response condition, $F_{1,92} = 11.42$, $p < 0.005$, $\eta_p^2 = 0.11$ and an interaction between response condition and congruency, $F_{1,92} = 6.47$, $p < 0.05$, $\eta_p^2 = 0.07$. There was also a two-way interaction between experiment and response condition, $F_{1,92} = 8.34$, $p < 0.01$, $\eta_p^2 = 0.08$, but the three-way interaction of experiment, response condition and congruency was not significant, $F_{1,92} = 3.39$, $p = 0.07$, $\eta_p^2 = 0.04$.

# 3. General discussion

As expected, Experiment 1 revealed a large congruency effect when an individual, isolated participant named both colours. Also as expected, Experiments 2 and 3 showed a congruency effect, albeit much smaller, when one participant named one or the other colour and the other participant did not name a colour (indicating that Stroop effects occur in go-no-go tasks). In Experiment 2, when partners did not provide feedback, the congruency effect was equivalent in both the single- and joint-response condition. However, Experiment 3 showed that when partners did provide feedback to each other, the congruency effect was larger in the joint-response than the single-response condition.

These findings indicate that, in the joint-response condition, a shared-Stroop effect occurs when participants are encouraged to monitor their partner's utterances for correctness. We suggest that the monitoring requirement means the participants tend to represent the partner's target utterance using the same mechanism that they use to represent their own target utterance (i.e. for self-monitoring). In solo Stroop tasks such as Experiment 1, speakers encounter interference from their own potential responses when they name a colour that does not match the word. In the shared-Stroop task reported in Experiment 3, participants appeared to consider their partner's potential responses in a way that interferes with their own responses (as indicated by the larger Stroop effect compared with the single-response condition). If one partner has to utter 'red' to the word *green* written in red, but knows that her partner regularly has to utter 'green' and is likely to prepare this response after seeing the word *green* written in red, she represents her partner's response and experiences increased interference as a result. This interfering representation could be a prediction of the partner's response, or it could be based on having comprehended the partner's response on preceding trials.

We assume that the small effect in the single-response condition (e.g. saying 'green' faster when responding to the word *green* than the word *red* written in green ink) occurred because participants still activate the inappropriate name (red) to some extent, even though they never produce that name in the experimental session. Alternatively, it may be related to semantic interference during the apprehension of the stimulus (see e.g. [45]), rather than at the response selection stage. Note that this finding is unlike what happens in the joint Simon effect, where the effect disappears in the go-no-go version of the task [17], possibly because in that case representing a co-acting partner is necessary to assign a spatial code to one's own response.

Instead, comprehending the colour word appears to automatically activate representations that are shared between the comprehension and the production system (e.g. semantic representations), thus causing a small amount of interference in the single-response condition as well. In the joint-response

condition, hearing one's partner produce the colour words does not, in and of itself, increase the activation level of production representations (see also [36]). But crucially, production-based representations are more likely to be activated when in addition participants have to monitor their partner's responses. Alternatively, it is possible that the monitoring task raises the activation level of comprehension representations that are used both in other- and in self-monitoring [14,42].

When we compare our task with other joint language tasks (e.g. picture naming in [8]), it appears that these representations may be affected by the nature of the task or the relationship between the speakers. In particular, when participants monitor each other's performance, representations of others' utterances are enhanced. By contrast, shared-Simon effects do not require participants to communicate their behaviour with their partners, to have any indication that their partner is responding, or even to be aware of their partner's presence (although the effects are sometimes enhanced when that is the case).

While Gambi et al. [8] found that people represent others' utterance even in language tasks that are not interactive, in that study there was no indication that speakers were representing what their partners were saying, but only that they were representing whether they were naming pictures or not. Participants in Kuhlen & Abdel Rahman [10], Baus et al. [4] and Demiral et al. [6] appeared to perform aspects of response preparation when it was their partner's turn to speak, but these participants were not simultaneously preparing to speak themselves. Further, in Baus et al. and Demiral et al. the partner was seated in the same room as the participant, while in Kuhlen and Abdel Rahman [10,33] robust evidence for joint interference was found only for co-present participants.

In this study, we showed that speakers experienced additional interference from simultaneously representing their own and their partner's target utterance, but only when doing so was required by the task. In other words, co-presence and acting alongside one another were not sufficient for the shared-Stroop effect; engaging in other-monitoring (a key component of linguistic interaction as observed in dialogue) was needed.

One interesting question relates to the degree to which our findings would extend to other joint language tasks. For example, would monitoring enhance interference in a joint PWI task in which the participants take turns to name pictures while ignoring distractor words, and those words are the names of pictures produced by their partner on different trials? While this situation is analogous to the current task—that is, irrelevant information is part of the partner's response set—participants may be less likely to strongly represent their partner's responses when there is a large set of them (as in PWI tasks). This might make it easier for participants to keep representations of self and other separate and reduce the demands on monitoring, thus also reducing the likelihood of finding that monitoring enhances joint interference.

Further, it is possible that interference only occurred in Experiment 3 (but not in Experiment 2) not because feedback enhanced representations of others' utterances, but rather because of the complexity of performing two tasks (naming and providing feedback). However, we argue that this conclusion is unlikely because the monitoring task was easy, as participants rarely made any errors. It could be argued that the monitoring task simply increased the saliency of the partner's responses, so that participants paid more attention to them, but this argument would not explain why the partner's utterances interfered with the participant's own utterances.

We instead propose that interference was enhanced because other-monitoring and self-monitoring rely on an overlapping set of mechanisms [11]. The need to produce one's own language and monitor the language of one's interlocutor is a hallmark of interactive language. Integrating linguistic representations for self- and other-produced utterances is most obviously relevant for interactive language, when interlocutors predict their partner's contributions and use these predictions to prepare [53] and time [54] their own utterances so that interlocutors do not extensively overlap or leave significant pauses. In other words, such representational integration may help promote the fluency of dialogue, as argued by Pickering & Garrod [16].

In sum, we have shown that participants sharing a Stroop task represent their partner's utterance using some of the same mechanisms they use to concurrently represent their own utterances, but only when the joint nature of the task is emphasized by the requirement to closely monitor one another's utterances. Thus, the integration of linguistic representations between self and other does take place, and it is more likely in more interactive situations.

Ethics. The research reported here received ethical approval from the Research Ethics Committee at the Department of Psychology, University of Edinburgh. Informed consent was obtained from all participants.
Data accessibility. All the experimental materials, data and analyses are available at https://osf.io/hg3du/.

Authors' contributions. M.J.P.: conceptualization, methodology, project administration, supervision, writing—original draft and writing—review and editing; J.F.M.: conceptualization, data curation, formal analysis, investigation, methodology, software, visualization, writing—original draft and writing—review and editing; C.G.: writing—original draft and writing—review and editing.

All authors gave final approval for publication and agreed to be held accountable for the work performed therein.

Conflict of interest declaration. We declare we have no competing interests.

Funding. M.J.P. and J.F.M. were supported by ESRC grant no. RES-062-23-0376. C.G. received no grants or funding in support of this project.

Acknowledgements. We thank Ciara Catchpole.

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
