## [Peer Review File · Royal Society Open Science]

Review History

RSOS-211143.R0 (Original submission)

Review form: Reviewer 1

Is the manuscript scientifically sound in its present form?

Yes

Are the interpretations and conclusions justified by the results?

Yes

Is the language acceptable?

Yes

Do you have any ethical concerns with this paper?

No

Have you any concerns about statistical analyses in this paper?

No

Recommendation?

Major revision is needed (please make suggestions in comments)

Comments to the Author(s)

This study investigates a social version of the Stroop task in which one partner is responsible for naming words of one color and the other partner is responsible for naming the other color. Previous studies suggest that subjects may not (or to a lesser extent) experience the classic Stroop interference when the second color is not responded to by either partner, but do experience interference when the partner responds to the second color. This effect speaks for participants representing the partner's task in a similar format as their own task. Current evidence is somewhat mixed though, in particularly regarding the social nature of this effect. The authors address the question whether Stroop interference elicited by the partner's task depends on the degree to which participants are required to monitor the partner's performance. Experiment 1 demonstrates the classic Stroop effect in a single-subject setting. Experiment 2 translates the Stroop task to the described social setting. Experiment 3 tests the same social setting but participants are asked to monitor the partner's response for correctness. Stroop interference is observed in all conditions. However, the Stroop effect seems overall smaller in a social setting and becomes larger only if participants are asked to monitor the partner's response. The authors conclude that participants monitor their partner's performance by representing their partner's responses in the same format as their own responses and this elicits interference.

The experiment is well designed, carefully conducted and the applied analyses approach seems appropriate. The manuscript is easy to follow and to the point. The predictions are laid out clearly and are related with expert knowledge of the field to the relevant literature (some very recent work may be worthwhile including, see below). A crucial analysis seems to be missing, though. Some theoretical assumptions the authors make would need to be connected more directly to the study, see comments below. The study contribute to the growing field of studies investigating joint action and the consequences joint action may have for language use. The study should definitely be made available to this community and is likely to be fruitful for developing further existing theoretical accounts.

One rather important part of the manuscript seems to be missing. The section "3.4 Combined analysis of Experiments 2 and 3" appears incomplete. I would have expected that in this section the authors report a pooled analysis of the data of Experiments 2 and 3. This analysis could test whether the interaction between social vs. single action reported in Exp. 3 hinges on the requirement to monitor the partner's response. This would be crucial evidence for the authors' proposal. On the other hand, if no differences between the experiments emerge the overall pattern seems to speak more for a robust Stroop effect independent of the partner's task (since the authors basically observe interference in all conditions). To address this it would also be interesting to pool the data of Experiment 1, as well, and directly compare the size of interference in the social and single version of the Stroop task.

There is a relevant set of studies that are currently published as CogSci conference proceedings by Miles Tufft and Daniel Richardson (2020) and as preprint by Anna Kuhlen and Rasha Abdel Rahman (2021). These studies both demonstrate that interference experienced in a picture-word interference task is substantially reduced (!) when the task is shared (one names the distractor the other the target). The authors propose that participants in this form of task splitting may offload one task to the partner or process the partner's task at a level leading to priming. Since the mechanisms behind picture-word interference are often related to Stroop interference these studies may be interesting to discuss.

<https://www.cognitivesciencesociety.org/cogsci20/papers/0152/0152.pdf>

<https://www.biorxiv.org/content/10.1101/2020.09.08.287458v2>

On a related note there is a recent publication by Kuhlen and Abdel Rahman (2021) that may be interesting to draw upon when discussing the role the social setting and social presence may play in eliciting interference by the partner's task:

<https://pubmed.ncbi.nlm.nih.gov/34292053/>

Further comments in chronological order:

Abstract:

I find it a bit confusing that there is no reference to Experiment 1 in the abstract. Could the authors foreshadow purpose, design and findings of Experiment 1, as well?

From the information given in abstract it is unclear what behavior pattern "interference" refers to. Could the authors briefly specify? (e.g., interference from the partner's task or interference from the ignored color)

Introduction:

Page 2, line 52ff: The authors write "There is increasing evidence that people also construct representations of their partner's utterances in joint language tasks [4-10]. But it is currently unclear to what extent representations of others' utterances and representation of one's own utterances are integrated and affect one another (i.e., to what extent they are shared)." I find the authors' description of what the literature has left unclear somewhat vague. I would find it very helpful if the authors be more specific on the open questions they identify in the literature.

Page 1, line 56ff: The authors write "This paper investigates whether the degree to which speakers construct shared representations of others' utterances depends on the nature of the task."

Is the word "shared" necessary here? I am somewhat confused about the term "shared representation" in general as I am not quite sure what exactly this means: the assumption that the task partner both represent the same action or the parity of representations between partner action and own action? Along similar lines, the authors write on page 3, line 31 "participants may construct individual and shared representations of joint tasks". What is the difference between individual and shared representations?

Page 3, lines 41-45: "Pickering and Garrod [11] proposed that other-monitoring and self-monitoring rely on similar mechanisms: in either case, the monitoring partner builds a predictive representation of either her own or the other's utterance, which is used to rapidly compare the expected utterance to the utterance that is actually produced, and either correct errors (in self-monitoring) or flag up a lack of understanding (in other-monitoring)."

On several occasions throughout the manuscript the authors state that participants predict their partner's naming response. This implies that it's not the partner's naming response itself that is triggering the interference, but instead the comparison between the anticipated (i.e. predicted) response and the actual response. Can the present study provide evidence that the effect is indeed based on a predictive mechanism, or could an alternative explanation be that the interfering representation is triggered by the partner's response (i.e. elicited by comprehension)?

Page 3, line 52: "Participants' naming latencies were affected by their beliefs about their partner": more accurately: beliefs about their partner's task, no?

Methods

Sample size:

- How did the authors determine the sample sizes?

- In Experiment 2 and 3 the description of the sample size relative to the distribution of the number of observations per condition was somewhat confusing. It took me a while to figure out how 72 recruited participants mapped onto 24 individual datasets entered for each condition. Could the authors provide some brief pointers to help the reader along?

General discussion

Page 8, line 10: "one participant named one or other colour ...". Is there a "the" missing here?

Review form: Reviewer 2 (Laurel Brehm)

Is the manuscript scientifically sound in its present form?

No

Are the interpretations and conclusions justified by the results?

No

Is the language acceptable?

No

Do you have any ethical concerns with this paper?

No

Have you any concerns about statistical analyses in this paper?

Yes

Recommendation?

Reject

Comments to the Author(s)

This manuscript reports three Stroop experiments in which individuals participated with or without a partner, and were required or not required to provide feedback about the partner's response. The data are interpreted under a language-as-joint-action framework as evidence for shared representations causing interference when the shared representations are made important. The experiments are simple and nicely designed, except for one caveat about sample size, but the framing does not at all set up the relevant background and is hard to follow. In particular, I am very concerned about the confounding of feedback versus monitoring. The paper also reads as sloppy: many topic sentences are missing in the intro and one section is entirely missing. My comments are below, beginning with more major comments, listed in descending order of importance.

1. Fundamentally, while the authors wish to be manipulating monitoring, what they are actually manipulating is the requirement to produce feedback. It's therefore important to be crystal clear in the introduction and discussion what the results do and do not support: can we actually know that providing feedback actually boosted monitoring? One alternate hypothesis is that perhaps providing feedback boosts activation at multiple levels of production planning, which reverberates through the system. I'm not sure this is a particularly parsimonious hypothesis, but it seems feasible. Please be clear what is manipulated, what is measured, and what it means--taking care to not confound monitoring and feedback as properties of the language system.

2. The introduction and general framing are hard to follow and do not support the study. As I see it, the basic claims that need support (because they are being tested) are (1) in joint

communication tasks such as joint Stroop, individuals form shared representations, (2) these joint representations are made stronger when necessary to perform the joint task (e.g., as in a feedback context), and (3), providing feedback encourages monitoring, which makes representations stronger.

As noted in the manuscript, the support for claims 1 and 2 is mixed: while there is some evidence for e.g. interference in production from representations of partner content, representations that are formed may only be partial, and representations of other actors may not actually be unique to a joint context. The evidence reviewed in total is sensible but a lot of findings are presented in somewhat of a jumbled order with relatively few topic sentences, which makes it hard to follow. There's a lot of space dedicated to non-linguistic joint action, which isn't super relevant and could be reduced. Then, the section on Stroop is hard to follow: it's hard to see the difference between the current results and Demiral et al., and hard to see how Saunders et al. connects at all (and I think that it does). Please make clearer what the prior work shows in terms of shared representations and the value added by running the current experiment.

Missing from the Introduction and General Discussion is a discussion of the inherent differences between picture naming and Stroop. It's not clear to me that the results would generalize to an even trivially more complex task like picture naming. Picture naming requires generating a unique label per trial (or sometimes repeating labels out of a moderately sized set), which requires lexical selection, which might/might not be modulated by inhibitory control, depending on your model. Stroop as operationalized here requires repeating two labels (reducing the formulation burden), but clearly has a big inhibitory control component. How might this difference impact language production? Would monitoring potentially play a bigger role in Stroop than in picture naming type tasks?

Finally, as mentioned in my first point, another topic largely missing from the Introduction is claim 3: feedback and how it relates to monitoring and interaction in general. The authors put a brief citation to one model of monitoring (line 44, p2), but this needs to be unpacked in more detail: it's an important component of the framework and it's not trivial. Note that there are many different monitoring mechanisms in models of language production. Some rely on production, and some rely on comprehension. Citing some other work on feedback and monitoring (in the Introduction and General Discussion) would help support the claims and provide the appropriate tempering when necessary. Please also define the terms self- and other-monitoring in the introduction to make it easier to follow the argument.

3. The sample sizes are all over the map. This makes the data seem fishy, especially since the authors are looking for the *absence* of an interaction. Please justify the sample size for each experiment. Why only 12 participants for Experiment 1, and why 72 for the other two?

4. Section 3.4 is missing: section 3.3 is instead duplicated.

5. a. The results are hard to follow: please specify where RT / accuracy is the dependent measure.
b. Given the very low error rates, I also wonder whether it is even appropriate to be examining accuracy at all: please justify that the number of observations is sufficiently large.
c. Were these repeated measures ANOVAs, tabulated over participants? If so, please state.

6. Your R analyses don't run. Files and variables are mis-named.

Smaller points:

1. There are some technical terms in the intro that are not clearly defined: go-nogo, P3 could be easily described in a few words when the terms are first presented, which would broaden the potential audience for the paper.

2. Errors in the monitoring task would be good to mention in a footnote or an additional row in Table 1.

Typos, etc:

page 4, line 60. "Responses were the time" This is referred to later as reaction time. Fix the definition.

p. 5, line 10. "in an individuals"

p. 6, line 6: "and but no effect"

Decision letter (RSOS-211143.R0)

Dear Dr Gambi

The Editors assigned to your paper RSOS-211143 "Interference in the Shared-Stroop Task: A Comparison of Self- and Other-Monitoring" have made a decision based on their reading of the paper and any comments received from reviewers.

Regrettably, in view of the reports received, the manuscript has been rejected in its current form. However, a new manuscript may be submitted which takes into consideration these comments.

We invite you to respond to the comments supplied below and prepare a resubmission of your manuscript. Below the referees' and Editors' comments (where applicable) we provide additional requirements. We provide guidance below to help you prepare your revision.

Please note that resubmitting your manuscript does not guarantee eventual acceptance, and we do not generally allow multiple rounds of revision and resubmission, so we urge you to make every effort to fully address all of the comments at this stage. If deemed necessary by the Editors, your manuscript will be sent back to one or more of the original reviewers for assessment. If the original reviewers are not available, we may invite new reviewers.

Please resubmit your revised manuscript and required files (see below) no later than 16-Mar-2022. Note: the ScholarOne system will 'lock' if resubmission is attempted on or after this deadline. If you do not think you will be able to meet this deadline, please contact the editorial office immediately.

Please note article processing charges apply to papers accepted for publication in Royal Society Open Science (<https://royalsocietypublishing.org/rsos/charges>). Charges will also apply to papers transferred to the journal from other Royal Society Publishing journals, as well as papers submitted as part of our collaboration with the Royal Society of Chemistry (<https://royalsocietypublishing.org/rsos/chemistry>). Fee waivers are available but must be requested when you submit your manuscript (<https://royalsocietypublishing.org/rsos/waivers>).

Thank you for submitting your manuscript to Royal Society Open Science and we look forward to receiving your resubmission. If you have any questions at all, please do not hesitate to get in touch.

on behalf of Dr Gina Grimshaw (Associate Editor) and Essi Viding (Subject Editor)
openscience@royalsociety.org

Associate Editor Comments to Author (Dr Gina Grimshaw):

Both reviewers find the experiments to be well-designed and analyses appropriate. I agree, and think that these studies, taken together, could provide a useful contribution to understanding of joint linguistic action. However, the manuscript is missing section 3.4 (currently this section just repeats Section 3.3), which provides the critical statistical comparison of Experiments 2 and 3. As a number of the study's conclusions hinge on this comparison, I recommend that the manuscript be rejected, but invite the authors to resubmit, including the relevant analyses. In addition, the reviewers raised a number of concerns about the theoretical framing of the research and specific mechanisms that are targeted in the experiments. These concerns (and other more minor recommendations) should also be addressed. The authors should also ensure that the R scripts run (e.g., do not rely on packages that might not be installed).

Reviewer comments to Author:

Reviewer: 1

Comments to the Author(s)

This study investigates a social version of the Stroop task in which one partner is responsible for naming words of one color and the other partner is responsible for naming the other color. Previous studies suggest that subjects may not (or to a lesser extent) experience the classic Stroop interference when the second color is not responded to by either partner, but do experience interference when the partner responds to the second color. This effect speaks for participants representing the partner's task in a similar format as their own task. Current evidence is somewhat mixed though, in particularly regarding the social nature of this effect. The authors address the question whether Stroop interference elicited by the partner's task depends on the degree to which participants are required to monitor the partner's performance. Experiment 1 demonstrates the classic Stroop effect in a single-subject setting. Experiment 2 translates the Stroop task to the described social setting. Experiment 3 tests the same social setting but participants are asked to monitor the partner's response for correctness. Stroop interference is observed in all conditions. However, the Stroop effect seems overall smaller in a social setting and becomes larger only if participants are asked to monitor the partner's response. The authors conclude that participants monitor their partner's performance by representing their partner's responses in the same format as their own responses and this elicits interference.

The experiment is well designed, carefully conducted and the applied analyses approach seems appropriate. The manuscript is easy to follow and to the point. The predictions are laid out clearly and are related with expert knowledge of the field to the relevant literature (some very recent work may be worthwhile including, see below). A crucial analysis seems to be missing, though. Some theoretical assumptions the authors make would need to be connected more directly to the study, see comments below. The study contribute to the growing field of studies investigating joint action and the consequences joint action may have for language use. The study should definitely be made available to this community and is likely to be fruitful for developing further existing theoretical accounts.

One rather important part of the manuscript seems to be missing. The section “3.4 Combined analysis of Experiments 2 and 3” appears incomplete. I would have expected that in this section the authors report a pooled analysis of the data of Experiments 2 and 3. This analysis could test whether the interaction between social vs. single action reported in Exp. 3 hinges on the requirement to monitor the partner’s response. This would be crucial evidence for the authors’ proposal. On the other hand, if no differences between the experiments emerge the overall pattern seems to speak more for a robust Stroop effect independent of the partner’s task (since the authors basically observe interference in all conditions). To address this it would also be interesting to pool the data of Experiment 1, as well, and directly compare the size of interference in the social and single version of the Stroop task.

There is a relevant set of studies that are currently published as CogSci conference proceedings by Miles Tufft and Daniel Richardson (2020) and as preprint by Anna Kuhlen and Rasha Abdel Rahman (2021). These studies both demonstrate that interference experienced in a picture-word interference task is substantially reduced (!) when the task is shared (one names the distractor the other the target). The authors propose that participants in this form of task splitting may offload one task to the partner or process the partner’s task at a level leading to priming. Since the mechanisms behind picture-word interference are often related to Stroop interference these studies may be interesting to discuss.

<https://www.cognitivesciencesociety.org/cogsci20/papers/0152/0152.pdf>
<https://www.biorxiv.org/content/10.1101/2020.09.08.287458v2>

On a related note there is a recent publication by Kuhlen and Abdel Rahman (2021) that may be interesting to draw upon when discussing the role the social setting and social presence may play in eliciting interference by the partner’s task:
<https://pubmed.ncbi.nlm.nih.gov/34292053/>

Further comments in chronological order:

Abstract:

I find it a bit confusing that there is no reference to Experiment 1 in the abstract. Could the authors foreshadow purpose, design and findings of Experiment 1, as well?

From the information given in abstract it is unclear what behavior pattern "interference" refers to. Could the authors briefly specify? (e.g., interference from the partner’s task or interference from the ignored color)

Introduction:

Page 2, line 52ff: The authors write “There is increasing evidence that people also construct representations of their partner’s utterances in joint language tasks [4-10]. But it is currently unclear to what extent representations of others’ utterances and representation of one’s own utterances are integrated and affect one another (i.e., to what extent they are shared).” I find the authors’ description of what the literature has left unclear somewhat vague. I would find it very helpful if the authors be more specific on the open questions they identify in the literature.

Page 1, line 56ff: The authors write “This paper investigates whether the degree to which speakers construct shared representations of others’ utterances depends on the nature of the task.”

Is the word “shared” necessary here? I am somewhat confused about the term “shared representation” in general as I am not quite sure what exactly this means: the assumption that the task partner both represent the same action or the parity of representations between partner action and own action? Along similar lines, the authors write on page 3, line 31 “participants may

construct individual and shared representations of joint tasks“. What is the difference between individual and shared representations?

Page 3, lines 41-45: “Pickering and Garrod [11] proposed that other-monitoring and self-monitoring rely on similar mechanisms: in either case, the monitoring partner builds a predictive representation of either her own or the other’s utterance, which is used to rapidly compare the expected utterance to the utterance that is actually produced, and either correct errors (in self-monitoring) or flag up a lack of understanding (in other-monitoring).”

On several occasions throughout the manuscript the authors state that participants predict their partner’s naming response. This implies that it’s not the partner’s naming response itself that is triggering the interference, but instead the comparison between the anticipated (i.e. predicted) response and the actual response. Can the present study provide evidence that the effect is indeed based on a predictive mechanism, or could an alternative explanation be that the interfering representation is triggered by the partner's response (i.e. elicited by comprehension)?

Page 3, line 52: “Participants’ naming latencies were affected by their beliefs about their partner”: more accurately: beliefs about their partner’s task, no?

Methods

Sample size:

- How did the authors determine the sample sizes?
- In Experiment 2 and 3 the description of the sample size relative to the distribution of the number of observations per condition was somewhat confusing. It took me a while to figure out how 72 recruited participants mapped onto 24 individual datasets entered for each condition. Could the authors provide some brief pointers to help the reader along?

General discussion

Page 8, line 10: “one participant named one or other colour ...”. Is there a “the” missing here?

Reviewer: 2

Comments to the Author(s)

This manuscript reports three Stroop experiments in which individuals participated with or without a partner, and were required or not required to provide feedback about the partner's response. The data are interpreted under a language-as-joint-action framework as evidence for shared representations causing interference when the shared representations are made important. The experiments are simple and nicely designed, except for one caveat about sample size, but the framing does not at all set up the relevant background and is hard to follow. In particular, I am very concerned about the confounding of feedback versus monitoring. The paper also reads as sloppy: many topic sentences are missing in the intro and one section is entirely missing. My comments are below, beginning with more major comments, listed in descending order of importance.

1. Fundamentally, while the authors wish to be manipulating monitoring, what they are actually manipulating is the requirement to produce feedback. It's therefore important to be crystal clear in the introduction and discussion what the results do and do not support: can we actually know that providing feedback actually boosted monitoring? One alternate hypothesis is that perhaps providing feedback boosts activation at multiple levels of production planning, which reverberates through the system. I'm not sure this is a particularly parsimonious hypothesis, but it seems feasible. Please be clear what is manipulated, what is measured, and what it means-- taking care to not confound monitoring and feedback as properties of the language system.

2. The introduction and general framing are hard to follow and do not support the study. As I see it, the basic claims that need support (because they are being tested) are (1) in joint communication tasks such as joint Stroop, individuals form shared representations, (2) these joint representations are made stronger when necessary to perform the joint task (e.g., as in a feedback context), and (3), providing feedback encourages monitoring, which makes representations stronger.

As noted in the manuscript, the support for claims 1 and 2 is mixed: while there is some evidence for e.g. interference in production from representations of partner content, representations that are formed may only be partial, and representations of other actors may not actually be unique to a joint context. The evidence reviewed in total is sensible but a lot of findings are presented in somewhat of a jumbled order with relatively few topic sentences, which makes it hard to follow. There's a lot of space dedicated to non-linguistic joint action, which isn't super relevant and could be reduced. Then, the section on Stroop is hard to follow: it's hard to see the difference between the current results and Demiral et al., and hard to see how Saunders et al. connects at all (and I think that it does). Please make clearer what the prior work shows in terms of shared representations and the value added by running the current experiment.

Missing from the Introduction and General Discussion is a discussion of the inherent differences between picture naming and Stroop. It's not clear to me that the results would generalize to an even trivially more complex task like picture naming. Picture naming requires generating a unique label per trial (or sometimes repeating labels out of a moderately sized set), which requires lexical selection, which might/might not be modulated by inhibitory control, depending on your model. Stroop as operationalized here requires repeating two labels (reducing the formulation burden), but clearly has a big inhibitory control component. How might this difference impact language production? Would monitoring potentially play a bigger role in Stroop than in picture naming type tasks?

Finally, as mentioned in my first point, another topic largely missing from the Introduction is claim 3: feedback and how it relates to monitoring and interaction in general. The authors put a brief citation to one model of monitoring (line 44, p2), but this needs to be unpacked in more detail: it's an important component of the framework and it's not trivial. Note that there are many different monitoring mechanisms in models of language production. Some rely on production, and some rely on comprehension. Citing some other work on feedback and monitoring (in the Introduction and General Discussion) would help support the claims and provide the appropriate tempering when necessary. Please also define the terms self- and other-monitoring in the introduction to make it easier to follow the argument.

3. The sample sizes are all over the map. This makes the data seem fishy, especially since the authors are looking for the *absence* of an interaction. Please justify the sample size for each experiment. Why only 12 participants for Experiment 1, and why 72 for the other two?

4. Section 3.4 is missing: section 3.3 is instead duplicated.

5. a. The results are hard to follow: please specify where RT / accuracy is the dependent measure.
 b. Given the very low error rates, I also wonder whether it is even appropriate to be examining accuracy at all: please justify that the number of observations is sufficiently large.
 c. Were these repeated measures ANOVAs, tabulated over participants? If so, please state.

6. Your R analyses don't run. Files and variables are mis-named.

Smaller points:

1. There are some technical terms in the intro that are not clearly defined: go-nogo, P3 could be easily described in a few words when the terms are first presented, which would broaden the potential audience for the paper.
2. Errors in the monitoring task would be good to mention in a footnote or an additional row in Table 1.

Typos, etc:

page 4, line 60. "Responses were the time" This is referred to later as reaction time. Fix the definition.

p. 5, line 10. "in an individuals"

p. 6, line 6: "and but no effect"

===PREPARING YOUR MANUSCRIPT===

===PREPARING YOUR REVISION IN SCHOLARONE===

Author's Response to Decision Letter for (RSOS-211143.R0)

See Appendix A.

Decision letter (RSOS-220107.R0)

Dear Dr Gambi

On behalf of the Editors, we are pleased to inform you that your Manuscript RSOS-220107 "Interference in the Shared-Stroop Task: A Comparison of Self- and Other-Monitoring" has been accepted for publication in Royal Society Open Science subject to minor revision in accordance with the referees' reports. Please find the referees' comments along with any feedback from the Editors below my signature.

Please submit your revised manuscript and required files (see below) no later than 7 days from today's (ie 22-Mar-2022) date. Note: the ScholarOne system will 'lock' if submission of the revision is attempted 7 or more days after the deadline. If you do not think you will be able to meet this deadline please contact the editorial office immediately.

on behalf of Dr Gina Grimshaw (Associate Editor) and Essi Viding (Subject Editor)
openscience@royalsociety.org

Associate Editor Comments to Author (Dr Gina Grimshaw):
Associate Editor
Comments to the Author:
Dear Dr Gambi and colleagues,

Thank you for revising your manuscript "Interference in the shared Stroop task: A comparison of self- and other-monitoring", and for your very detailed response to reviewers. I have carefully read the manuscript and your response myself, and have not needed to send it back to reviewers. I find that you have addressed the reviewers' comments well, and think that the manuscript is

clearer for your efforts. I therefore recommend acceptance of the manuscript pending some very minor revisions.

1. I had a little trouble with your R script and needed to rename the first column to make it run. I recommend that you check this before publication.
2. p. 14 line 23 - please define the go-nogo task in terms of the Stroop task (I think readers understand that go-nogo means you respond on some trials but not others; but in a Stroop context, presumably this means that they respond to some colours and not others?).
3. p. 15, line 38 - you suggest that participants in the joint-Stroop task "took turns responding". Most people would interpret this to mean that they alternated. Since your stimuli are presumably randomised, this isn't entirely accurate. Or is it? (i.e., was there a constraint that the responder alternated from trial to trial. Please reword so that it is clear.

===PREPARING YOUR MANUSCRIPT===

one version should clearly identify all the changes that have been made (for instance, in coloured highlight, in bold text, or tracked changes);

===PREPARING YOUR REVISION IN SCHOLARONE===

-- If you are requesting an article processing charge waiver, you must select the relevant waiver option (if requesting a discretionary waiver, the form should have been uploaded, see 'File upload' above).

-- If you have uploaded any electronic supplementary (ESM) files, please ensure you follow the guidance at <https://royalsociety.org/journals/authors/author-guidelines/#supplementary-material> to include a suitable title and informative caption. An example of appropriate titling and captioning may be found at https://figshare.com/articles/Table_S2_from_Is_there_a_trade-off_between_peak_performance_and_performance_breadth_across_temperatures_for_aerobic_scope_in_teleost_fishes_/3843624.

Author's Response to Decision Letter for (RSOS-220107.R0)

See Appendix B.

Decision letter (RSOS-220107.R1)

Dear Dr Gambi,

I am pleased to inform you that your manuscript entitled "Interference in the Shared-Stroop Task: A Comparison of Self- and Other-Monitoring" is now accepted for publication in Royal Society Open Science.

on behalf of Dr Gina Grimshaw (Associate Editor) and Essi Viding (Subject Editor)
openscience@royalsociety.org

Appendix A

Dear Dr. Grimshaw,

Thank you for the opportunity of submitting a revised version of our manuscript, *Interference in the Shared-Stroop Task: A Comparison of Self- and Other-Monitoring*.

First of all, we would like to apologise for the missing section (3.4) in the previous version of this manuscript. It was due to a copy and paste error while transferring our manuscript to the RSOS template. This error has now been fixed and section 3.4 reports combined analyses for Experiment 2 and 3.

We have addressed all of the reviewers' concerns regarding the theoretical framing of our manuscript, as detailed in our point-by-point response below.

Finally, an updated version of the R script has been uploaded. This includes the missing *plyr* package. All the scripts should run smoothly.

Yours sincerely,

**Martin Pickering
Janet McLean
Chiara Gambi**

Point-by-point response

Reviewer: 1

Comments to the Author(s)

This study investigates a social version of the Stroop task in which one partner is responsible for naming words of one colour and the other partner is responsible for naming the other colour. Previous studies suggest that subjects may not (or to a lesser extent) experience the classic Stroop interference when the second colour is not responded to by either partner, but do experience interference when the partner responds to the second colour. This effect speaks for participants representing the partner's task in a similar format as their own task. Current evidence is somewhat mixed though, in particularly regarding the social nature of this effect. The authors address the question whether Stroop interference elicited by the partner's task depends on the degree to which participants are required to monitor the partner's performance. Experiment 1 demonstrates the classic Stroop effect in a single-subject setting. Experiment 2 translates the Stroop task to the described social setting. Experiment 3 tests the same social setting but participants are asked to monitor the partner's response for correctness. Stroop interference is observed in all conditions. However, the Stroop effect seems overall smaller in a social setting and becomes larger only if participants are asked to monitor the partner's response. The authors conclude that participants monitor their partner's performance by representing their partner's responses in the same format as their own responses and this elicits interference.

The experiment is well designed, carefully conducted and the applied analyses

approach seems appropriate. The manuscript is easy to follow and to the point. The predictions are laid out clearly and are related with expert knowledge of the field to the relevant literature (some very recent work may be worthwhile including, see below). A crucial analysis seems to be missing, though. Some theoretical assumptions the authors make would need to be connected more directly to the study, see comments below. The study contribute to the growing field of studies investigating joint action and the consequences joint action may have for language use. The study should definitely be made available to this community and is likely to be fruitful for developing further existing theoretical accounts.

Thank you for the positive appraisal of our work!

One rather important part of the manuscript seems to be missing. The section “3.4 Combined analysis of Experiments 2 and 3” appears incomplete. I would have expected that in this section the authors report a pooled analysis of the data of Experiments 2 and 3. This analysis could test whether the interaction between social vs. single action reported in Exp. 3 hinges on the requirement to monitor the partner’s response. This would be crucial evidence for the authors’ proposal. On the other hand, if no differences between the experiments emerge the overall pattern seems to speak more for a robust Stroop effect independent of the partner’s task (since the authors basically observe interference in all conditions).

We apologies for the missing crucial comparison between Experiments 2 and 3 – we inadvertently omitted it from the manuscript when transferring it to the RSOS template prior to submission. This is now reported in section 3.4, and it confirms that the difference between the congruency effect in the single- and joint-response conditions was greater in Experiment 3 than Experiment 2 (i.e., a three-way interaction of congruency, condition and experiment for the response times analysis). Thus, the presence of partner feedback in Experiment 3 did increase the congruency effect that occurred in the joint-response condition compared to the experiment without feedback.

To address this it would also be interesting to pool the data of Experiment 1, as well, and directly compare the size of interference in the social and single version of the Stroop task.

Thanks for this suggestion but we do not believe that a direct comparison between the solo experiment (Exp 1) and the joint experiments (Exp 2 and 3) is informative. Exp 1 was a standard two-colour Stroop task carried out by individual participants responding to both colours. In contrast, in Exp 2 and 3 participants took part in pairs and always responded to only one colour, while we varied (i) whether their partner also responded to the other colour (within-experiment) and (ii) whether they provided feedback to their partner (between-experiments). Thus, there were multiple factors that differed between Exp 1 and the other two experiments (presence of another person, number of colours the participants responded to) and we would not be able to tease apart which of these factors was responsible for differences in the size of the congruency effect. This is precisely the reason why we compared the single to the joint response conditions within experiments 2 and 3. Note also that the number of participants in

Experiment 1 was different from the number of participants in each of the other two experiments, and each participant provided double the number of trials, which also complicates the comparison.

There is a relevant set of studies that are currently published as CogSci conference proceedings by Miles Tufft and Daniel Richardson (2020) and as preprint by Anna Kuhlen and Rasha Abdel Rahman (2021). These studies both demonstrate that interference experienced in a picture-word interference task is substantially reduced (!) when the task is shared (one names the distractor the other the target). The authors propose that participants in this form of task splitting may offload one task to the partner or process the partner's task at a level leading to priming. Since the mechanisms behind picture-word interference are often related to Stroop interference these studies may be interesting to discuss.

<https://www.cognitivesciencesociety.org/cogsci20/papers/0152/0152.pdf>
<https://www.biorxiv.org/content/10.1101/2020.09.08.287458v2>

Many thanks for these suggestions. We now cite both studies on p. 4, in relation to related evidence for a reduction in perceptual conflict in the shared Stroop task of Demiral et al. (2016).

On a related note there is a recent publication by Kuhlen and Abdel Rahman (2021) that may be interesting to draw upon when discussing the role the social setting and social presence may play in eliciting interference by the partner's task:

<https://pubmed.ncbi.nlm.nih.gov/34292053/>

Many thanks for this suggestion. This study is now mentioned on p. 3.

Further comments in chronological order:

Abstract:

I find it a bit confusing that there is no reference to Experiment 1 in the abstract. Could the authors foreshadow purpose, design and findings of Experiment 1, as well?

From the information given in abstract it is unclear what behavior pattern "interference" refers to. Could the authors briefly specify? (e.g., interference from the partner's task or interference from the ignored colour)

Thanks for these suggestions. We now mention Experiment 1 and specify what is meant by interference in the abstract.

Introduction:

Page 2, line 52ff: The authors write "There is increasing evidence that people also construct representations of their partner's utterances in joint language tasks [4-10]. But it is currently unclear to what extent representations of others' utterances and representation of one's own utterances are integrated and affect one another (i.e., to what extent they are shared)." I find the authors' description of what the literature has left unclear somewhat vague. I would find it very helpful if the authors be more specific on the open questions they identify in the literature.

We have now clarified this, by adding the following: “Specifically, some studies suggest that representations of others’ utterances have effects that are analogous to those of representations of one’s own utterances (e.g., interacting in similar ways with other linguistic representations or eliciting similar brain responses; [4,6,10]); but other work suggests that representation of others’ utterances are not as tightly integrated with representations of one’s own utterances [5,7-9].” (p. 2).

Page 1, line 56ff: The authors write “This paper investigates whether the degree to which speakers construct shared representations of others’ utterances depends on the nature of the task.”

Is the word “shared” necessary here? I am somewhat confused about the term “shared representation” in general as I am not quite sure what exactly this means: the assumption that the task partner both represent the same action or the parity of representations between partner action and own action? Along similar lines, the authors write on page 3, line 31 “participants may construct individual and shared representations of joint tasks“. What is the difference between individual and shared representations?

Thanks for raising this important point. We recognise that we used the word “shared” somewhat loosely in the previous version. We have now replaced every instance of this word with a more explicit formulation (except when referring to the shared Stroop task, where it serves as a convenient short-hand): what we mean is that partners represent each other’s actions/utterances in addition to their own and that these representations of self and other are tightly integrated so they influence each other in measurable ways.

Page 3, lines 41-45: “Pickering and Garrod [11] proposed that other-monitoring and self-monitoring rely on similar mechanisms: in either case, the monitoring partner builds a predictive representation of either her own or the other’s utterance, which is used to rapidly compare the expected utterance to the utterance that is actually produced, and either correct errors (in self-monitoring) or flag up a lack of understanding (in other-monitoring).”

On several occasions throughout the manuscript the authors state that participants predict their partner’s naming response. This implies that it’s not the partner’s naming response itself that is triggering the interference, but instead the comparison between the anticipated (i.e. predicted) response and the actual response. Can the present study provide evidence that the effect is indeed based on a predictive mechanism, or could an alternative explanation be that the interfering representation is triggered by the partner’s response (i.e. elicited by comprehension)?

This study cannot adjudicate between a prediction-based and a comprehension-based account. Accordingly, we have played down our references to prediction throughout the manuscript. We briefly mention the two alternatives in the GD on p. 8. Note however that on each trial in our task only one participant responds, so if the increased interference effect is due to comprehension of the partner’s response, then this would be comprehension of that response on previous trials. The prediction-based alternative is that the interference is caused by prediction

of the partner's response on subsequent trials. In any case, that prediction would likely be based on having observed the partner's response on previous trials.

Page 3, line 52: "Participants' naming latencies were affected by their beliefs about their partner": more accurately: beliefs about their partner's task, no?

Corrected, thanks.

Methods

Sample size:

- How did the authors determine the sample sizes?

We have now included explicit justifications of our chosen sample sizes for all 3 experiments (see the respective Participants sections). We did not conduct a formal power calculation before data collection (and we now say so explicitly).

- In Experiment 2 and 3 the description of the sample size relative to the distribution of the number of observations per condition was somewhat confusing. It took me a while to figure out how 72 recruited participants mapped onto 24 individual datasets entered for each condition. Could the authors provide some brief pointers to help the reader along?

In section 3.2.2, we now write: "The design yielded data for 24 single response participants (either red or green) and 24 joint response participants (both participants in each of the 12 joint pairs provided data, one red one green).", which hopefully makes it more explicit.

General discussion

Page 8, line 10: "one participant named one or other colour ...". Is there a "the" missing here?

Corrected, thanks

Reviewer: 2

Comments to the Author(s)

This manuscript reports three Stroop experiments in which individuals participated with or without a partner, and were required or not required to provide feedback about the partner's response. The data are interpreted under a language-as-joint-action framework as evidence for shared representations causing interference when the shared representations are made important. The experiments are simple and nicely designed, except for one caveat about sample size, but the framing does not at all set up the relevant background and is hard to follow. In particular, I am very concerned about the confounding of feedback versus monitoring. The paper also reads as sloppy: many topic sentences are missing in the intro and one section is entirely missing. My comments are below, beginning with more major comments, listed in descending order of importance.

1. Fundamentally, while the authors wish to be manipulating monitoring, what they are actually manipulating is the requirement to produce feedback. It's therefore important to be crystal clear in the introduction and discussion what the results do and do not

support: can we actually know that providing feedback actually boosted monitoring? One alternate hypothesis is that perhaps providing feedback boosts activation at multiple levels of production planning, which reverberates through the system. I'm not sure this is a particularly parsimonious hypothesis, but it seems feasible. Please be clear what is manipulated, what is measured, and what it means-- taking care to not confound monitoring and feedback as properties of the language system.

We agree this is an important clarification. We have added this sentence on p. 4: "While we do not know the exact mechanism by which receiving (or providing) feedback affects language processing, it is likely that introducing a requirement to provide feedback will make participants more likely to closely monitor their partners' utterances."

2. The introduction and general framing are hard to follow and do not support the study. As I see it, the basic claims that need support (because they are being tested) are (1) in joint communication tasks such as joint Stroop, individuals form shared representations, (2) these joint representations are made stronger when necessary to perform the joint task (e.g., as in a feedback context), and (3), providing feedback encourages monitoring, which makes representations stronger.

As noted in the manuscript, the support for claims 1 and 2 is mixed: while there is some evidence for e.g. interference in production from representations of partner content, representations that are formed may only be partial, and representations of other actors may not actually be unique to a joint context. The evidence reviewed in total is sensible but a lot of findings are presented in somewhat of a jumbled order with relatively few topic sentences, which makes it hard to follow.

There's a lot of space dedicated to non-linguistic joint action, which isn't super relevant and could be reduced. Then, the section on Stroop is hard to follow: it's hard to see the difference between the current results and Demiral et al., and hard to see how Saunders et al. connects at all (and I think that it does). Please make clearer what the prior work shows in terms of shared representations and the value added by running the current experiment.

Many thanks for prompting us to streamline and rationalise the review of the literature. The picture that emerges from previous work using Stroop and related joint language paradigms is indeed complex. In the new, extensively revised introduction, we present these findings in a different order and take care to clarify what previous work shows and what the current study crucially adds. We have also significantly reduced the space devoted to non-linguistic joint action.

Missing from the Introduction and General Discussion is a discussion of the inherent differences between picture naming and Stroop. It's not clear to me that the results would generalize to an even trivially more complex task like picture naming. Picture naming requires generating a unique label per trial (or sometimes repeating labels out of a moderately sized set), which requires lexical selection, which might/might not be modulated by inhibitory control, depending on your model. Stroop as operationalized here requires repeating two labels (reducing the formulation burden), but clearly has a big inhibitory control component. How might this difference impact language production? Would monitoring potentially play a bigger role in Stroop than in picture naming type tasks?

Many thanks – this is a valuable suggestion. In the introduction (p. 4), we caution the reader that comparing joint Stroop and joint picture naming/PWI tasks may be problematic. We list some key methodological differences. Further, in the GD (p. 9), we discuss whether our findings would extend to joint picture naming/PWI tasks given they likely place different demands on monitoring, as suggested by the review.

Finally, as mentioned in my first point, another topic largely missing from the Introduction is claim 3: feedback and how it relates to monitoring and interaction in general. The authors put a brief citation to one model of monitoring (line 44, p2), but this needs to be unpacked in more detail: it's an important component of the framework and it's not trivial. Note that there are many different monitoring mechanisms in models of language production. Some rely on production, and some rely on comprehension. Citing some other work on feedback and monitoring (in the Introduction and General Discussion) would help support the claims and provide the appropriate tempering when necessary. Please also define the terms self- and other-monitoring in the introduction to make it easier to follow the argument.

We have added several references to other major monitoring frameworks (Intro, p. 2; Levelt, 1983; Nozari et al., 2011; Van de Meerendonk et al., 2009 and the recent review by Gauvin and Hartsuiker, 2020). We also acknowledge that the representations we show to be integrated in our task could either be production representations or comprehension representations (GD, p. 8). Finally, we now define self- and other-monitoring on first mention (p. 2) as suggested.

3. The sample sizes are all over the map. This makes the data seem fishy, especially since the authors are looking for the *absence* of an interaction. Please justify the sample size for each experiment. Why only 12 participants for Experiment 1, and why 72 for the other two?

The larger sample size for Exp 2 and Exp 3 is justified by their design. In these experiments, we varied Condition (single vs. joint-response) between participants, and 24 participants provided data in each condition. In addition, note that while in the joint-response condition both participants in a pair provided data (so it was sufficient to test 12 pairs to get data from 24 participants) in the single-response condition, only one participant per pair did, so in order to obtain data for 24 participants we had to test 24 pairs = 48 participants. In total, we thus tested $48+24 = 72$ participants per experiment. In contrast, Experiment 1 tested participants individually: only 12 participants were tested because Stroop interference is known to have a large effect size (Olsson-Collentine et al., 2020). We instead expected the go-nogo version of the task to show reduced interference because the interfering colour was now not in the response set (Sharma & McKenna, 1998), and we also expected the presence of a partner to reduce interference (Klauer et al., 2008), so we increased the sample size.

4. Section 3.4 is missing: section 3.3 is instead duplicated.

Apologies for this oversight. This has now been rectified.

5. a. The results are hard to follow: please specify where RT / accuracy is the dependent measure.

In sections 3.2.2 (Experiment 2), 3.3.2 (Experiment 3) and 3.4 (combined analysis) we added paragraph labels (Response Times, Accuracy) to make clear which set of analyses we are describing in each paragraph.

b. Given the very low error rates, I also wonder whether it is even appropriate to be examining accuracy at all: please justify that the number of observations is sufficiently large.

While the error rate was indeed small (See Table 1), it was comparable to that found in some experiments that are especially designed to elicit speech errors. For example, in Exp 3, pairs with feedback, joint-response condition, we recorded 55 errors out of 120 trials*12 pairs = 1440 responses, which corresponds to an error rate of 4%. In comparison, Nootboom & Quené (2020; Exp 1) reported 963 errors/6784 responses = 14% and Nootboom & Quené (2019; Exp 1) reported 824/16454 = 5%. Both of those experiments were designed to elicit errors. Importantly, we had many observations (trials) per cell of the design, so although the numbers are small we have no reason to suspect our estimates to be inaccurate.

c. Were these repeated measures ANOVAs, tabulated over participants? If so, please state.

We now specify on p. 5: “Response times and accuracy were analysed using mixed-design ANOVAs, after averaging over trials by participants.”

6. Your R analyses don't run. Files and variables are mis-named.

Apologies – we have now re-checked the script and it should run. A package was missing (*pylr*) and there was a mismatch between the name of the data file for Experiment 1 in the script vs. the folder.

Smaller points:

1. There are some technical terms in the intro that are not clearly defined: go-nogo, P3 could be easily described in a few words when the terms are first presented, which would broaden the potential audience for the paper.

Thanks for the suggestion, we now define these terms on their first occurrence.

2. Errors in the monitoring task would be good to mention in a footnote or an additional row in Table 1.

Thanks for this suggestion. As we mention in the paper (p. 9), there were hardly any errors at all in the monitoring task, suggesting it was easy for participants to check whether their partner's response was correct or not (possibly because the

number of errors in the Stroop task was low, see the reviewer's previous comment). This statement is based on observations by the second author, who attended all sessions. However, because the numbers were so low, we did not deem it necessary to verify this by listening to the audio files that recorded the feedback response. This means that, unfortunately, we do not have the exact figures at hand.

Typos, etc:

page 4, line 60. "Responses were the time" This is referred to later as reaction time. Fix the definition.

We consistently refer to this DV as response times. This sentence has been

amended: "Response times were the time taken to state the colour of the word from its onset."

p. 5, line 10. "in an individuals"

Corrected.

p. 6, line 6: "and but no effect"

Corrected

References

Gauvin, H. S., Hartsuiker, R. J. 2020 Towards a new model of verbal monitoring. *Journal of Cognition*. 3, 1.

Klauer, K. C., Herfordt, J., Voss, A. (2008) Social presence effects on the stroop task: Boundary conditions and an alternative account. *Journal of Experimental Social Psychology*. 44, 469-476.

Levelt, W. J. 1983 Monitoring and self-repair in speech. *Cognition*. 14, 41-104.

Nooteboom, S., & Quené, H. (2019). Temporal aspects of self-monitoring for speech errors. *Journal of Memory and Language*, 105, 43-59.

Nooteboom, S. G., & Quené, H. (2020). Repairing speech errors: Competition as a source of repairs. *Journal of Memory and Language*, 111, 104069.

Nozari, N., Dell, G. S., Schwartz, M. F. 2011 Is comprehension necessary for error detection? A conflict-based account of monitoring in speech production. *Cognitive Psychology*. 63, 1-33.

Olsson-Collentine, A., Wicherts, J. M., & van Assen, M. A. (2020). Heterogeneity in direct replications in psychology and its association with effect size. *Psychological Bulletin*, 146(10), 922.

Sharma, D., & McKenna, F. P. (1998). Differential components of the manual and vocal Stroop tasks. *Memory & Cognition*, 26(5), 1033-1040.

Van de Meerendonk, N., Kolk, H. H., Chwilla, D. J., Vissers, C. T. W. 2009 Monitoring in language perception. *Language and linguistics Compass*. 3, 1211-1224.

Appendix B

Dear Dr. Grimshaw,

Thank you for your positive evaluation of our revised manuscript, *Interference in the Shared-Stroop Task: A Comparison of Self- and Other-Monitoring*.

Below we briefly list how we addressed the remaining minor points you highlighted.

1. I had a little trouble with your R script and needed to rename the first column to make it run. I recommend that you check this before publication.

Thank you for spotting this issue. We have carefully checked the R script and associated data files and made sure the analysis script now runs without returning any errors. We think the problem may have been that we had updated the R script but not the data files.

2. p. 14 line 23 - please define the go-nogo task in terms of the Stroop task (I think readers understand that go-nogo means you respond on some trials but not others; but in a Stroop context, presumably this means that they respond to some colours and not others?).

This sentence now reads (changes in bold): “Demiral et al. [6] had participants perform a delayed go-nogo version of the Stroop task, where participants responded on trials **where the word was printed in one ink colour (e.g., green; go trials)** but not on trials **where the word was printed in a different ink colour (e.g., red; nogo trials)**.”

2. p. 15, line 38 - you suggest that participants in the joint-Stroop task "took turns responding". Most people would interpret this to mean that they alternated. Since your stimuli are presumably randomised, this isn't entirely accurate. Or is it? (i.e., was there a constraint that the responder alternated from trial to trial. Please reword so that it is clear.

This sentence now reads (changes in bold): “conditions in which another person was present and **responded on trials that were nogo trials for the participant** (joint-response task).” **As you correctly noted, the participants did not necessarily alternate – stimuli were randomised.**

Yours sincerely,

Martin Pickering
Janet McLean
Chiara Gambi